# Cooccurring Type 1 Diabetes Mellitus and Autoimmune Thyroiditis in a Girl with Craniofrontonasal Syndrome: Are *EFNB1* Variants Associated with Autoimmunity?

**DOI:** 10.3390/ph15121535

**Published:** 2022-12-10

**Authors:** Sebla Güneş, Jiangping Wu, Berk Özyılmaz, Reyhan Deveci Sevim, Tolga Ünüvar, Ahmet Anık

**Affiliations:** 1Division of Pediatric Endocrinology, Department of Pediatrics, Faculty of Medicine, Aydın Adnan Menderes University, 09100 Aydın, Turkey; 2Centre de Recherche, Centre Hospitalier de l’Université de Montréal (CHUM), Montreal, QU H2X 0A9, Canada; 3Genetic Diagnosis Center, Tepecik Training and Research Hospital, University of Health Sciences, 35020 Izmir, Turkey

**Keywords:** craniofrontonasal syndrome, ephrin B1, autoimmune diseases, type 1 diabetes mellitus, autoimmune thyroiditis

## Abstract

Craniofrontonasal syndrome (CFNS), also known as craniofrontonasal dysplasia, is an X-linked inherited developmental malformation caused by mutations in the ephrin B1 (*EFNB1*) gene. The main phenotypic features of the syndrome are coronal synostosis, hypertelorism, bifid nasal tip, dry and curly hair, and longitudinal splitting of nails. A 9-year-and-11-month-old girl with CFNS was admitted due to polyuria, polydipsia, fatigue, and abdominal pain. On physical examination, she had the classical phenotypical features of CFNS. Genetic tests revealed a c.429_430insT (p.Gly144TrpfsTer31) heterozygote variant in the *EFNB1* coding region. The patient was diagnosed with type 1 diabetes mellitus (T1DM) and autoimmune thyroiditis based on laboratory findings and symptoms. The mother of the patient, who had the same CFNS phenotype and EFNB1 variant, was screened for autoimmune diseases and was also with autoimmune thyroiditis. This is the first report describing the association of CFNS with T1DM and autoimmune thyroiditis in patients with *EFNB1* mutation.

## 1. Introduction

Craniofrontonasal syndrome (CFNS), also known as craniofrontonasal dysplasia, is an X-linked inherited developmental malformation and was described for the first time by Cohen in 1979 [1]. The main features of the phenotype are coronal synostosis, hypertelorism, bifid nasal tip, dry and curly hair, and longitudinal splitting of the nails. Most of the patients with CFNS are females [2].

Ephrin B1 (EFNB1), which is coded by the X-linked *EFNB1* gene, is a transmembrane ligand that interacts with EPH kinase (erythropoietin-producing hepatocellular carcinoma receptor tyrosine kinase) B subfamily members. Compagni et al. first reported that EFNB1 controls skeletal pattern formation, and its deletion in mice causes a phenotype reminiscent of CFNS, including a female-biased incidence in spite of *Efnb1* being X-linked [3]. Based on this clue, several groups quickly validated that *EFNB1* mutations are associated with human CFNS [4,5]. The causes of the paradoxical female bias for this X-linked CFNS have been investigated. Three possible explanations, which are not mutually exclusive, backed by some clinical and experimental evidence, are proposed. A. De novo germline EFNB1 mutation in the X allele of males can only be transmitted to female offspring and thus causes CFNS in females. B. EFNB1 is vital for embryonic development, and most males with EFNB1 mutations were aborted. C. Due to postzygotic X-inactivation, females with hemizygous EFNB1 mutations are mosaic. Such mosaicism increases the survival of the inflicted females as the *EFNB1* in some patches of cells is from the wild-type allele, but at the same time, it brings about the often-observed varied clinical manifestations due to the unpredictable nature of mosaicism. 

The roles of EPHs and EFNs in immunity have been studied by many research groups [6,7]. As our current study is about EFNB1, we provide an EFNB1-focused review below. EFNB1 is implicated in many aspects of immune responses. In the T cell compartment, it is necessary for proper thymocyte development and the proliferation, homeostatic expansion, survival, differentiation, chemotaxis, and function of mature T cells [8,9,10,11,12,13,14,15]. EFNB1 generally has positive effects on these processes. In consistence with this notion, EFNB1 in T cells is overexpressed locally in the brain lesions of multiple sclerosis patients or systemically in the peripheral blood of rheumatoid arthritis patients [12,13]. In the B cell compartment, EFNB1 expression in the germinal center (GC) inhibits the retention of follicular T-helper cells (Tfh) in the GC and reduces interaction between B cells and Tfh. As a consequence, the B cell maturation is compromised [16]. However, this negative effect is counteracted by EFNB1’s positive effect on enhancing IL-21 production by Tfh cells, with IL-21 being a critical cytokine for GC B cell proliferation and maturation. The final outcome of the humoral immune response seems to depend on the balance between these two opposing effects. In adipose cells, EFNB1 deletion promotes the production of inflammatory cytokine MCP-1 production by monocytes [17]. Therefore, EFNB1 has cell- and tissue-dependent positive or negative effects on immune and inflammatory responses. The final effect might depend on the balance of these two opposing forces. 

In animal models or patients, it is difficult to find the overall effects of EFNB1 in the immune system due to the vital role of EFNB1 in fetal development and organ functions. The complete inactivation of this gene often causes lethality [18,19], preventing us from assessing the long-term outcome of EFNB1 deletion. Hence, so far, there is no report on abnormal immune responses in patients with EFNB1 mutations. 

In this study, we, for the first time, report the cooccurrence of CFNS and autoimmune diseases in a family with *EFNB1* mutation. 

## 2. Clinical Report

### 2.1. A General Description of the Patients

A 9-year-and-11-month-old girl diagnosed with CFNS (heterozygote mutation in *EFNB1* gene c.429_430insT (p.Gly144TrpfsTer31) was admitted due to polyuria, polydipsia, fatigue, and abdominal pain. Her past medical history showed that she was born at the 32nd gestational week by cesarean section with a birth weight of 2650 g. She had a bifid big toe on her left foot. Two proximal and two distal phalanxes were seen in the left big toe in her pre-operative direct radiography (Figure 1A). She underwent surgery for the bifid big toe when she was eight years and 11 months old. Bilateral clinodactyly of the 3rd, 4th, and 5th toes was observed (Figure 1B).

There was no consanguinity between the parents. On physical examination, her weight was 24 Kg (−1.54 SDS (standard deviation score)), height was 136 cm (+0.18 SDS), body mass index was 12.98 kg/m^2^ (−2.25 SDS), and head circumference was 49 cm (−2.44 SDS). She had hypertelorism, a wide and flattened nose base, bilateral epicanthus, strabismus in the right eye (sursoadductorius), a bifid nasal tip, a flat nasal bridge, dry and curly hair, low-set ears, brachycephaly, a short-webbed neck, and low hairline (Figure 2). Nails on both hands and feet were concave with longitudinal splits. A pubertal examination showed breasts in Tanner’s stage 1 and pubic hair in stage 2. Palpation of the thyroid gland revealed soft and non-tender thyromegaly.

Her 47-year-old mother had the same phenotypical features, as well as the *EFNB1* mutation compatible with the CFNS diagnosis. On physical examination, her weight was 63 kg, her height was 166.2 cm, with a body mass index of 22.8 kg/m^2^ and a head circumference of 51.7 cm. She had hypertelorism, a wide and flattened nose base, bilateral epicanthus, strabismus in the left eye (sursoadductorius), a bifid nasal tip, a flat nasal bridge, dry and wavy hair, low-set ears and brachycephaly (Figure 3). Nails on both hands and feet were concave with longitudinal splits and deformation. Palpation of the thyroid gland revealed soft and non-tender thyromegaly.

### 2.2. Face Anthropometry 

An anthropometric measurement of the girl patient was conducted according to Van den Elzen et al. [2], with the following parameters registered and indices calculated. Bi-ocular width: 5 cm; craniofacial length: 22 cm; face width: 13 cm; intercanthal width: 6 cm; nasal length: 4.5 cm; nasal tip protrusion: 1.5 cm; length of the upper half face: 6.5 cm; upper lip length: 2 cm; intercanthal index (intercanthal length/bi-ocular width): 0.4; index of the upper half face (the length of the upper half face/face width): 0.5; nasal tip protrusion-nasal length index (nasal tip protrusion-nasal length): 0.33; nasal/craniofacial length index (nasal length/craniofacial length): 0.20; index of the nasal-upper half face (nasal length/length of the upper half face): 0.69; upper lip-upper half face index (upper lip length-upper half face length): 0.30. 

### 2.3. Laboratory Findings

The girl patient was diagnosed with diabetic ketoacidosis according to the following laboratory findings: venous glucose of 449 mg/dl; ketonuria at pH 7.07 and HCO3 of 8.2 mmol/L. She was also diagnosed with type 1 diabetes mellitus (T1DM) based on the following findings: glycated hemoglobin (HbA1c), 11.2%; C-peptide, 0.51 ng/mL; anti-insulin antibody, 28.5 IU/mL (N (normal): 0–20 IU/mL); anti-glutamic acid decarboxylase antibody, 35.63 IU/mL (N: 0–17 IU/mL); and positive anti-islet antibody (N: negative). 

Complete blood count, serum electrolytes, liver transaminases, kidney function tests, and lipid profile were all in the normal range. Autoimmune thyroiditis was diagnosed according to the following parameters: serum-free thyroxin (fT4), 0.9 ng/dL (N: 1–1.7 ng/dL); anti-thyroglobulin antibody >1000 IU/mL (N: <1); and anti-thyroid peroxidase (TPO) antibody, >1000 IU/mL (N: <20 IU/mL). Thyroid ultrasonography showed an enlarged thyroid gland (thyroid volume: 12.86 mL (+6.32 SDS)] with a diffusely heterogeneous coarse echotexture and multiple discrete hypoechoic micronodules. Thyroid-stimulating hormone (1.5 mIU/L (N: 0.6–4.8 mIU/L)) and serum IgA (1.64 g/L (N: 0.45–2.85)) were in the normal range. Tissue anti-transglutaminase IgA and anti-endomysium IgA antibodies were negative.

The mother had normal fT4 and TSH levels but abnormally high titers of anti-thyroglobulin and anti-thyroid peroxidase antibodies (13.96 IU/mL (N: <4) and 488 IU/mL (N: <9), respectively). She was diagnosed with euthyroid autoimmune thyroiditis.

### 2.4. Genetic Tests 

Chromosome analysis from peripheral blood lymphocytes of the girl patient showed 23 pairs of chromosomes, with XX for the 23rd pair. Sanger sequencing of the *EFNB1* gene revealed a c.429_430insT (p.Gly144TrpfsTer31) heterozygote variant (Figure 4). This variant was not registered in databases. As a result, it was interpreted as “Pathogenic” according to ACMG criteria (PVS1, PM2). The variant was creating an insertion of a T base in the position of 68059530, which was causing a frameshift and production of a termination codon after 31 bases. Her mother also had the same heterozygote *EFNB1* variant according to Sanger sequencing (Figure 4).

### 2.5. Screening for Comorbidities 

The angle of the clavicles with the sternum of the girl patient was wider than normal (Figure 5). The X-ray of the cranium was normal.

The right kidney was located in the pelvis with anteriorly rotated hilus and localized calyceal enlargement (7 × 12 mm). There was no significant hydronephrosis. The left kidney was placed mildly inferiorly to the expected localization with a double collective system and grade 1 hydronephrosis of the upper pole of a double collective system. Tc-99m dimercaptosuccinic acid scan showed irregular uptake in the left kidney and hypoactive parenchymal areas (scar) in the right kidney. 

An ophthalmologic examination revealed hypermetropia, astigmatism, amblyopia, and hypertropia. Hearing tests, neuropsychiatric and cognitive tests, electrocardiography, and echocardiography were all normal. 

## 3. Discussion 

Ephrin B1, which is coded by the *EFNB1* gene in the X chromosome, is a type I membrane protein and a ligand of EPH receptor tyrosine kinases. It plays an essential role in cell migration, repulsion, and adhesion during neuronal, vascular, and epithelial development. X-linked mutation of this gene causes CFNS. In the girl patient described in this study, typical phenotypic features of CFNS, such as hypertelorism, bifid nasal tip, dry and curly hair, and bumpy nails with longitudinal splitting, were present. Also, strabismus, low-set ears, polydactyly (bifid big toe), and clinodactyly, which are other possible phenotypical features of CFNS, were manifested. The angle of the clavicles with the sternum was wide, and facial anthropometric measurements were abnormal; these phenotypes are compatible with CFNS, although nystagmus and cardiac anomalies, which can be associated with the syndrome, were not found. Overall, the clinical manifestations of the girl patient are typical of CFNS. The fact that the patient presented some but not all of the CFNS phenotypes are compatible with previous reports that this syndrome has quite varied manifestations, which are probably caused by postzygotic X-inactivation of *EFNB1* randomly in different tissues due to random chimerism. This girl patient had T1DM and autoimmune thyroiditis. Her mother, who similarly suffered from CFNS with the same EFNB1 variant, was also diagnosed with autoimmune thyroiditis. The association of these autoimmune diseases with CFNS was not previously reported, either due to overlooking or that these autoimmune diseases are only present in some patients whose *EFNB1* mutation in some mosaic tissues affects immune regulation.

EPH kinases are the largest family of receptor protein tyrosine kinases [8]. The ligands of these kinases are EFNs, which are also cell surface molecules. The kinases and ligands can send signals in both directions, i.e., canonic forward signaling from EFN ligands to receptor EPH kinases and reverse signaling from EPH receptors to EFN ligands. Some EPHs and EFNs are expressed in the immune cells, such as T cells, B cells, and monocytes/macrophages [6,8,16,20,21,22]. The forward signaling from EFNB1 through EPH receptors enhances peripheral T-cell proliferation, cytokine production, and cytotoxic activity [8]. For thymocytes, such forward signaling protects thymocytes at different differentiation stages from apoptosis [10]. Lou et al. [13] reported that animals with T cell-specific double deletion of EFNB1 and EFNB2 have defective proliferation and differentiation of Th1 and Th17 cells. Thus, EFNB1 and EFNB2 play an important role in the optimal maturation of Th1 and Th17 cells, which have a proven critical role in causing multiple sclerosis [23]. In another study, the same researchers showed that mouses with EFNB1 and EFNB2 mutations exhibit an insufficient immune response against lymphocytic choriomeningitis virus infection, in part due to reduced IL-6 stimulation, which has a significant effect on viral immunity [15]. These findings reveal that EFNB1 expressed in the T cell compartment has a positive role in immune regulation, which is at odds with the occurrence of autoimmune diseases in patients with a loss-of-function mutation.

On the other hand, EFNB1 expression in some other cell types does have a constitutive negative impact on the immune responses. As reviewed in the Introduction, EFNB1 in GC B cells repels the Tfh and hence compromises plasma cell maturation and antibody production [16]. EFNB1 expression in adipose cells suppresses inflammatory cytokine secretion by monocytes [17] (PLoS ONE 2013, T Mori, 8:e76199). Such inhibitory effects of EFNB1, if revoked, can probably disturb the normal check-and-balance in the immune system and increase the risk of autoimmune diseases. 

T1DM is mainly a T-cell-mediated autoimmune disease caused by immune destruction of the β-cells [24]. This destruction is believed to be triggered by one or more ambient agents in genetically susceptible individuals who stay asymptomatic for months or years [25]. Although T-cells are the major players in T1DM, B-cell-mediated humoral immunity is also involved in its pathogenesis [26]. Indeed, the anti-CD20 antibody, a B cell-depleting biologic drug, is highly promising in treating T1DM in both humans and mice [27,28].

CD4 T cells play a major role in the pathogenesis of autoimmune thyroiditis [29], but thyroid-specific antibodies also contribute to the pathogenesis of the disease [30]. Anti-CD20 Ab is also effective in treating such thyroiditis [31].

Considering the documented inhibitory effect of EFNB1 on the humoral immune responses and the contributing roles of B cells and autoantibodies to the pathogenesis of T1DM and autoimmune thyroiditis, it is tempting to speculate that a putative loss of EFNB1-mediated inhibition of humoral immune responses increases the risks of these autoimmune diseases. Of course, EFNB1 mutations can cause autoimmune diseases in multiple other ways, as this molecule is widely expressed in different immune cells. The concurrence of T1DM and autoimmune thyroiditis is a reminiscence of autoimmune polyendocrine syndrome type I (APSI) caused by compromised T cell negative selection in the thymus due to thymic epithelial *AIRE* mutations [32]. EFNB1 is highly expressed in the thymus, and it will be interesting to investigate whether its mutation affects the negative selection in the thymus. Due to the vital importance of EFNB1, we cannot create animal models with EFNB1 being deleted in all the cell types. Similarly, humans with such deletion probably die during fetal development. Consequently, we cannot study the overall outcome of EFNB1 deletion for the immune system. The current two CFNS cases have provided us with a unique opportunity to assess the overall and long-term outcome of EFNB1 mutation in the immune system. These patients have survived and presented CFNS and autoimmune diseases, probably due to the chimerism, in which some but not all of their tissues have dysfunctional EFNB1. The normal cells without the mutation allowed them to survive, and the tissues with the mutation cause CFNS and autoimmune diseases. We hope that the reporting of these cases can lead to additional studies linking EFNB1 mutations to autoimmune diseases, which might have been previously overlooked in CFNS patients. 

Due to EFNB1’s vital functions, indiscriminately using EFNB1 as a drug target will be too lethal. Our current study suggests that EFNB1 signaling (either the forward signaling from EFNB1 to EPH receptor kinases or reverse signaling from EPHs to EFNB1) is implicated in the pathogenesis of T1DM and autoimmune thyroiditis. Therefore, some signaling molecules in these pathways in certain types of cells might be valid and novel drug targets for such autoimmune diseases, which are widespread and are not restricted to patients with EFNB1 mutations. Future investigations elucidating the cell types and signaling molecules implicated in the autoimmune diseases caused by EFNB1 mutation can provide more information about these potential drug targets. 

In conclusion, we discovered a previously unknown association of CFNS with T1DM and autoimmune thyroiditis. We encourage physicians to be vigilant to the possible concurrent autoimmune disorders during the follow-up of CFNS patients with *EFNB1* mutations. Moreover, some molecules in the EFNB1 signaling pathways might be potential drug targets for these autoimmune diseases. 

## Figures and Tables

**Figure 1 pharmaceuticals-15-01535-f001:**
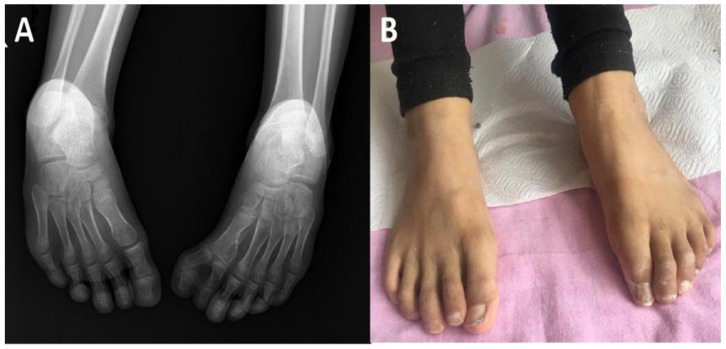
The abnormal feet of the girl patient before and after the surgery. (**A**) Direct radiography of the left bifid big toe with two proximal and two distal phalanxes before the surgery. (**B**) A photo of the feet after the surgery. Longitudinal splitting in the toenails is remarkable.

**Figure 2 pharmaceuticals-15-01535-f002:**
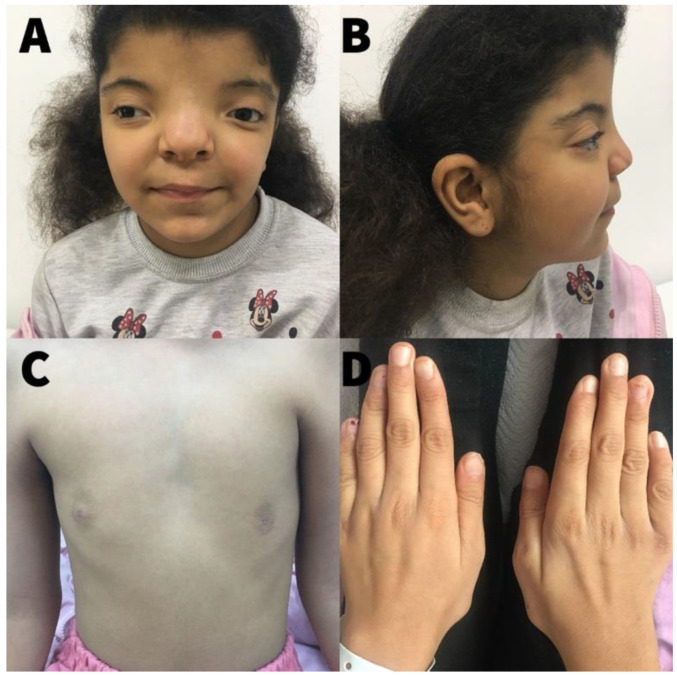
The abnormal appearance of the girl patient. (**A**) Syndromic facial appearance of the girl patient. (**B**) Brachycephaly and low-set ear. (**C**) Abnormal areolar leveling. (**D**) Longitudinal bumps and splitting on fingernails.

**Figure 3 pharmaceuticals-15-01535-f003:**
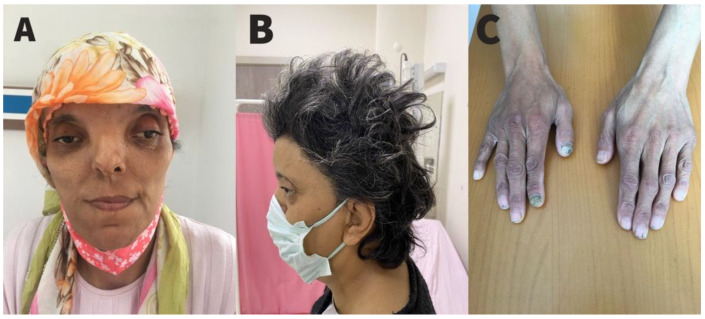
The abnormal appearance of the mother. (**A**) Syndromic facial appearance of the mother. (**B**) Brachycephaly and low-set ear. (**C**) Longitudinal bumps, splitting, and deformation on fingernails.

**Figure 4 pharmaceuticals-15-01535-f004:**
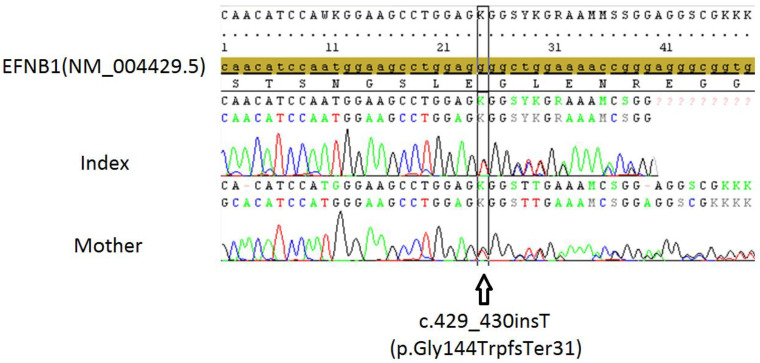
EFNB1 gene sanger sequencing of the patient (index) and her mother. The sequencing of the EFNB1 gene revealed c.429_430insT (p.Gly144TrpfsTer31) heterozygote variant. The variant created an insertion of a T base in the position of 68059530, which caused a frameshift and production of a termination codon after 31 bases.

**Figure 5 pharmaceuticals-15-01535-f005:**
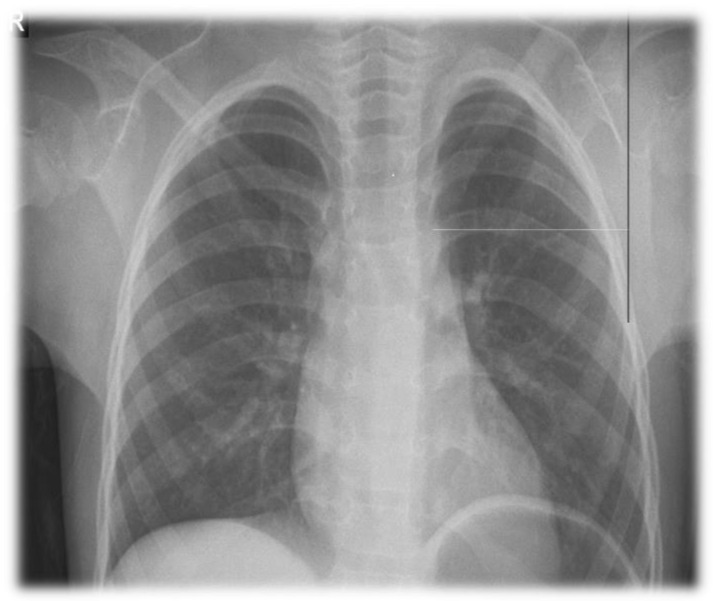
Chest X-ray findings of the patient. The angle of the clavicles with the sternum is wider than normal.

## Data Availability

Data is contained within the article.

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
