# Peer review of "Cooccurring Type 1 Diabetes Mellitus and Autoimmune Thyroiditis in a Girl with Craniofrontonasal Syndrome: Are EFNB1 Variants Associated with Autoimmunity?"

_pharmaceuticals, 2022, doi:10.3390/ph15121535_

Round 1

Reviewer 1 Report

The present manuscript titled 'Cooccurring Type 1 Diabetes Mellitus and Autoimmune Thyroiditis in a Girl with Craniofrontonasal Syndrome: Are EFNB1 Variants Associated with Autoimmunity? is very well written with all aspects clearly explained but the literature cited is not recent so i would suggest to update the literature.

Author Response

Response to Reviewer 1 Comments

Point 1: The present manuscript titled 'Cooccurring Type 1 Diabetes Mellitus and Autoimmune Thyroiditis in a Girl with Craniofrontonasal Syndrome: Are EFNB1 Variants Associated with Autoimmunity? is very well written with all aspects clearly explained but the literature cited is not recent so I would suggest to update the literature.

Response 1: Our literature review focuses on EFNB1’s role in immune responses and the information related to T1DM and auto-immune thyroiditis, particularly the humoral immune responses in these two diseases, as autoantibodies were detected in our patients. To the best of our knowledge, these reviews are up to date. Of course, there are recent reports and reviews about the roles of other members of EPH/EFN in the immune system, although they are less relevant to our topic. With that said, we have now added a couple several recent review articles that provide an updated overview of the EPH/EFN in the immune system.

Reviewer 2 Report

Page 2  para 2

“There was no consanguinity between the parents. On physical examination, her weight was 24 kg (-1.54 SDS (standard deviation score)”

Change to 2.4 Kg

Page 7, para 3

“T1DM is mainly a T-cell-mediated autoimmune disease caused by immune destruction of the –cells”

Insert Beta cells

Page 7 para 6: 1st 3 lines

CFNS is a very serious hereditary disease, and there is no cure. The best remedy is prenatal genetic screening followed by abortion, which is the protocol for many severe congenital disorders such as Down's syndrome.”

I object to the authors recommending abortion in this manuscript. The aim of the manuscript is to describe co- association of  EFNB1 gene  mutation with a certain phenotype and auto-immune problems in a mother and daughter. The recommendation for abortion is beyond the scope of this paper’s aim.   These 3 lines should be deleted.

Author Response

Response to Reviewer 2 Comments

-Point 1 (Page 2  para): There was no consanguinity between the parents. On physical examination, her weight was 24 kg (-1.54 SDS (standard deviation score)”

Change to 2.4 Kg

Response 1: Our patient's weight is 24 kilograms; so we can not understand why you request us to change it to 2.4 Kg.

-Point 2 (Page 7, para 3): “T1DM is mainly a T-cell-mediated autoimmune disease caused by immune destruction of the –cells”

Insert Beta cells

Response 2: The Greek letter ß was probably lost during file conversion. The typo has been corrected.

-Point 3 (Page 7 para 6: 1st 3 lines): “CFNS is a very serious hereditary disease, and there is no cure. The best remedy is prenatal genetic screening followed by abortion, which is the protocol for many severe congenital disorders such as Down's syndrome.”

I object to the authors recommending abortion in this manuscript. The aim of the manuscript is to describe co- association of  EFNB1 gene mutation with a certain phenotype and auto-immune problems in a mother and daughter. The recommendation for abortion is beyond the scope of this paper’s aim.   These 3 lines should be deleted.

Response 3: This is probably a politically sensitive issue, although this is the only sensible thing to do for the well-being of future patients and families. We have removed this suggestion in the text per the reviewer’s suggestion.